# SimCSE++: Improving Contrastive Learning for Sentence Embeddings from Two Perspectives [*]

**Jiahao Xu[1]   Wei Shao[2]   Lihui Chen[1]   Lemao Liu[3]**

[1]Nanyang Technological University, [2]City Univeristy of Hong Kong, [3]Tencent AI Lab

[1]jiahao004@e.ntu.edu.sg    elhchen@ntu.edu.sg
[2]weishao4-c@my.cityu.edu.hk
[3]redmondliu@tencent.com

## Abstract

This paper improves contrastive learning for sentence embeddings from two perspectives: handling dropout noise and addressing feature corruption. Specifically, for the first perspective, we identify that the dropout noise from negative pairs affects the model's performance. Therefore, we propose a simple yet effective method to deal with such type of noise. Secondly, we pinpoint the rank bottleneck of current solutions to feature corruption and propose a dimension-wise contrastive learning objective to address this issue. Both proposed methods are generic and can be applied to any contrastive learning based models for sentence embeddings. Experimental results on standard benchmarks demonstrate that combining both proposed methods leads to a gain of 1.8 points compared to the strong baseline SimCSE configured with BERT base. Furthermore, applying the proposed method to DiffCSE, another strong contrastive learning based baseline, results in a gain of 1.4 points.

## 1 Introduction

Sentence representation, which transforms sentence semantic information from discrete language space into dense vectors, is one of the most fundamental tasks in natural language processing, as it serves as the central role for a wide range of downstream applications (e.g., information retrieval, semantic comparison, question answering, and language translation). Sentence representation has been constantly evolving (Pennington et al., 2014; Zhang et al., 2020; Carlsson et al., 2021), and it achieves even stronger performance when utilizing pre-trained language models (PLM) (Devlin et al., 2019; Delobelle et al., 2020). Moreover, on top of PLMs, a number of post-processing strategies achieve even better performance. For example, Li et al. (2020) employs a flow-based model and Su

et al. (2021) applies the whitening process to flatten a uniform distribution of representations.

More recently, remarkable advancements have been achieved by contrastive learning (CL) on sentence embeddings (Gao et al., 2021), which cleverly makes use of *dropout randomness* (Bouthillier et al., 2015) to construct positive pairs in an unsupervised way. Since then, many notable variants have been proposed under the contrastive learning framework to intensify performance by constructing hard contrastive pairs (Giorgi et al., 2021; Kim et al., 2021; Yan et al., 2021; Wu et al., 2022; Zhang et al., 2022b; Chuang et al., 2020), introducing other CL-based objectives (Zhang et al., 2021, 2022c; Chuang et al., 2022; Zhang et al., 2022a; Tan et al., 2022; Xu et al., 2023) or utilizing more sophisticated similarity metrics (Zhang et al., 2022c).

In this paper, we improve the sentence embedding models from two perspectives: dropout noise and feature corruption. Specifically, first, we empirically study the effects of dropout randomness on positive pairs and negative pairs in the CL-based objective. We find that modest dropout noise in the positive pairs is beneficial to the model performance whereas dropout noise in negative pairs is harmful. We provide an explanation from the principle of noise contrastive estimation (Gutmann and Hyvärinen, 2012) and the role of dropout in constructing positive pairs. Based on these findings, we propose a simple yet effective strategy, *off-dropout*, which turns off the dropout randomness in negative pairs to further improve the performance.

Second, we revisit the issue of feature corruption on the sentence embedding and empirically study the well-known solution recently proposed by Zbontar et al. (2021); Klein and Nabi (2022) to this problem. Surprisingly, we find that this solution does not improve performance under the contrastive learning framework for sentence embeddings. We further analyze this finding and identify

[*]The source code is available at https://github.com/Jiahao004/SimCSE-plus-plus.

the reason behind it as the rank bottleneck issue in the mini-batch embedding matrix. To tackle this issue, we propose a simple *dimension-wise contrastive learning* (DCL) to break down the bottleneck, which eventually enhances the baseline performance.

As a result, by combining the proposed off-dropout and DCL, we have advanced the SimCSE baseline by 1.9 points. Furthermore, our reproduced results have shown that we advanced the current state-of-the-art model, DiffCSE (Chuang et al., 2022), by 1.4 points.

In general, our contribution is three-fold:

1. We, for the first time, point out that dropout noise from negative pairs has a side effect on model performance and propose an off-sampling strategy to alleviate this side effect.

2. We identify the rank bottleneck in the current solution to the feature corruption problem and propose a novel dimension-wise CL objective to avoid the bottleneck.

3. Experimental results on standard benchmarks for sentence embeddings show that the combination of our proposed methods outperforms strong baselines by a margin and achieves a new state-of-the-art.

## 2 Related Work

### 2.1 Sentence Representation

Early studies for sentence representations leverage the word2vec (Mikolov et al.) ideas. Semantic information can be captured by predicting a sentence from its surrounding sentences (Kiros et al., 2015; Hill et al., 2016; Logeswaran and Lee, 2018). Pagliardini et al. (2018) aggregates the n-gram embeddings using a pooling strategy, which achieves a strong result. With the development of large-scale pre-trained language models (Devlin et al., 2019; Liu et al., 2020), sentence representation methods begin to utilize PLMs' strong language representation ability. For example, Reimers and Gurevych (2019) employs siamese network with PLMs for supervised sentence representation, while Li et al. (2020) and Su et al. (2021) apply post-processing on top of PLM's representations.

Recent studies on sentence embeddings are based on the strong baseline SimCSE (Gao et al., 2021). Under the SimCSE framework, several studies focus on constructing hard contrastive pairs:

Zhang et al. (2020) utilize all the output token representations, Yan et al. (2021) enhance dropout augmentation, Giorgi et al. (2021) employ context sentences and Kim et al. (2021) contrast each layer representation within PLMs. Some studies aim to counter the PLMs bias towards sentence representations: Carlsson et al. (2021) employ heterogeneous model structure, Zhou et al. (2022) filter out noise from negatives. Others introduce more effective contrastive learning framework: Chuang et al. (2022) introduce ELECTRA (Clark et al., 2020) with equivariant contrastive learning (Dangovski et al., 2021), Zhang et al. (2022c) utilize ArcFace (Deng et al., 2019) framework.

All the previous studies on sentence embeddings have concentrated on developing more intricate frameworks based on the SimCSE framework. These advancements include creating more efficient training samples, introducing advanced metrics, and incorporating additional training tasks. In contrast to these existing studies, our research aims to enhance the contrastive learning framework itself. Specifically, we address two issues: the problem of dropout noise in the representation and the feature corruption caused by the correlation between different dimensions of the representation.

### 2.2 Contrastive Learning and NCE

The importance of contrastive learning has long been recognized. In NLP research fields, contrastive learning is introduced into sentence representations (Giorgi et al., 2021; Wu et al., 2020), text classification (Fang et al., 2020), information extraction (Qin et al., 2021), machine translations (Pan et al., 2021), question answering (Karpukhin et al., 2020) etc.

The concept of contrastive learning is based on Noise Contrastive Estimation (NCE) (Gutmann and Hyvärinen, 2010), which involves maximizing the probability of target signals by comparing them with randomly sampled noise. While NCE uses nonlinear logistic regression to distinguish between observed data and artificially generated noise using the log-density function, contrastive learning utilizes InfoNCE (Oord et al., 2018) objectives to discriminate between positive similarities and similarities among negative samples within the batch.

The previous research on NCE and contrastive learning primarily concentrates on the noise arising from the sampling of negative examples. However, this study investigates the noise originating

from dropout randomness and examines the impact of dropout randomness on sentence embeddings, considering both negative and positive examples.

## 2.3 Feature Corruption Issue

Feature corruption is a non-trivial problem in representation learning, where each dimension of the model shares high similarities with others. This issue hinders the expressive capacity to convey complex information effectively, as the diversity of each dimension value is constrained by such correlation.

Several studies (Li et al., 2020; Su et al., 2021) have attempted to address this issue by achieving a more independent embedding space through post-processing. However, as demonstrated in Wang et al. (2022), these post-processing methods primarily enhance performance for sentence pairs with low similarity and fail to improve performance for pairs with high similarity.

Recently, Zbontar et al. (2021) proposed BarlowTwins as a solution for such a issue in images. Inspired by the redundancy-reduction principle of neuroscientist H. Barlow, BarlowTwins minimizes redundancy between different dimensions, naturally reducing similarity across each dimension. Unlike post-processing methods, this approach addresses the problem in an end-to-end manner. Furthermore, a direct application of BarlowTwins on sentence embeddings (Klein and Nabi, 2022) achieves comparable performance to SimCSE.

In contrast to previous research that simply applies the BarlowTwins objective to the SimCSE framework, our study investigates the rank bottleneck issue of BarlowTwins in the context of sentence representation. We tackle this issue and improve the model's performance accordingly.

## 3 Improving Dropout Noise in CL

SimCSE framework plays a central role in recent sentence embedding strategies. It is a simple contrastive learning framework that learns by identifying positive pairs among in-batch negatives. Specifically, for a given sentence $x_i$, let $f(\cdot)$ denotes a pre-trained language model, and it is used to generate two views $(z_i^1, z_i^2)$ of the identical sentences $x_i$ via different dropout patterns:

$$
\begin{aligned}
z_i^1 &= f(x_i; \xi_i^1) \\
z_i^2 &= f(x_i; \xi_i^2)
\end{aligned}
\tag{1}
$$

where $\xi_i^1$ and $\xi_i^2$ denote two samples from the dropout random variable $\xi$ (Srivastava et al., 2014).

SimCSE (Gao et al., 2021) aims to maximize the agreement between positive pairs $\langle z_i^1, z_i^2 \rangle$ and minimize the $N-1$ in-batch negatives $\langle z_i^1, z_j^2 \rangle$ using the InfoNCE objective (Oord et al., 2018):

$$
\ell_{\text{Info}}^i = -\log \frac{e^{s(z_i^1, z_i^2)}}{e^{s(z_i^1, z_i^2)} + \sum_{j=1, j \neq i}^{N} e^{s(z_i^1, z_j^2)}}
\tag{2}
$$

Here, $s(\cdot, \cdot)$ is the similarity measure between two inputs (i.e., $cos\_sim(\cdot, \cdot)/\tau$, where $\tau$ is the temperature). In Equation (2), $\langle z_i^1, z_j^2 \rangle$ is a negative pair, and the dropout random variable $\xi$ is used as an augmentation function for positive pairs, i.e., $\langle z_i^1, z_i^2 \rangle$.

### 3.1 Dropout Noise in Negative Estimation

**Empirical study on dropout noise**  In PLMs such as BERT, it is shown that dropout plays an important role in training because of the regularization effect. In CL-based sentence embeddings, the training objective Eq. (2) involves $2 \times N$ BERT structures, and thus the role of dropout in Eq. (2) might be more complex. This motivates us to study the effect of dropout.

As presented in Eq. (1), dropout is determined by the random variable $\xi$ and thus $z_i^1$ (or $z_i^2$) ($i \in [1, N]$) contains some noise due to the random variable $\xi$. To study the effect of dropout noise, we respectively add more noise (**+Noise**) or reduce some noise (**-Noise**) to $z_i^1$ (or $z_i^2$) and then study their final performance.

Specifically, to introduce more noise to $z_i^1$ (or $z_i^2$), we add a small Gaussian noise as follows:

$$
\begin{aligned}
z_i^{1,+} &= f(x_i; \xi_i^1) + g^1 \\
z_i^{2,+} &= f(x_i; \xi_i^2) + g^2
\end{aligned}
$$

Where $g^1$ and $g^2$ are Gaussian with the mean $0$ and variance $0.1$. On the other hand, according to the Central Limit Theorem (Fischer), the $K$ sample average converges to its expectation with $1/K$ of the original variance [1]. Therefore, to reduce the noise from $z_i^1$ (or $z_i^2$), we could simply use the following mean sampling:

$$
\begin{aligned}
z_i^{1,-} &= \frac{1}{K} \sum_{k=1}^{K} f(x_i; \xi_i^{1,k}) \\
z_i^{2,-} &= \frac{1}{K} \sum_{k=1}^{K} f(x_i; \xi_i^{2,k})
\end{aligned}
$$

---

[1]$K = 10$ in this paper.

| Method | | STS12 | STS13 | STS14 | STS15 | STS16 | STS-B | SICK-R | Avg. |
|---|---|---|---|---|---|---|---|---|---|
| SimCSE | | 68.40 | 82.41 | 74.38 | 80.91 | 78.56 | 76.85 | **72.23** | 76.25 |
| Pos. | +Noise | 68.59 | 82.57 | 73.26 | 80.79 | 76.72 | 75.56 | 69.24 | 75.25 |
| | -Noise | 55.22 | 71.78 | 60.55 | 70.79 | 72.95 | 64.58 | 63.00 | 65.55 |
| Neg. | +Noise | 65.13 | 82.45 | 71.69 | 79.67 | 77.73 | 75.03 | 70.04 | 74.53 |
| | -Noise | **70.25** | **83.73** | **75.61** | **82.25** | **78.77** | **77.58** | 71.26 | **77.06** |

Table 1: Performance on STS benchmark with adding/reducing noise in positive and negative pairs.

where $\xi_i^{1,k}$ and $\xi_i^{2,k}$ are independently sampled from the dropout variable $\xi$, and thus $z_i^{1,\text{-}}$ contains less noise than $z_i^1$.

**Experimental results and findings** Since Eq. (2) contains positive pair $\langle z_i^1, z_i^2 \rangle$ and negative pair $\langle z_i^1, z_j^2 \rangle$, we individually conduct experiments to estimate the impact of the noise in positive and negative pairs. Respectively, SimCSE+Pos+Noise is achieved by replacing the positive pair $s(z_i^1, z_i^2)$ by $s(z_i^{1,+}, z_i^{2,+})$ in Eq. (2), and Sim-CSE+Neg+Noise is achieved by replacing the negative pair $s(z_i^1, z_j^2)$ by $s(z_i^{1,+}, z_j^{2,+})$ in Eq. (2). Similary, SimCSE+Pos-Noise applies $s(z_i^{1,\text{-}}, z_i^{2,\text{-}})$ as the replacement of positive pair $s(z_i^1, z_i^2)$ and SimCSE+Neg-Noise uses $s(z_i^{1,\text{-}}, z_j^{2,\text{-}})$ to replace negative pair $s(z_i^1, z_j^2)$.

Table 1 shows that increasing the noise level for both positive and negative embeddings may degenerate the performance while reducing the noise level for negative embeddings is helpful for model performance. In summary, we can obtain the following findings: 1) having modest noise in positive pairs is necessary to make CL successful and reducing noise in positive pairs is harmful to the performance; 2) the model performance is related to the noise level of negative pairs: more noise degrades the performance while less noise improves the performance.

**Theoretical Explanation** Contrastive learning compares the similarity of positive examples with negative ones. This idea is based on Noise Contrastive Estimation (NCE) (Gutmann and Hyvärinen, 2010), where the positive similarity score is the target signal that NCE tries to maximize, while the negative similarity score is the corresponding noise signal.

The InfoNCE loss in Eq. (2) follows Noise Contrastive Estimation (NCE) (Gutmann and Hyvärinen, 2010). It shows that the model converges faster and performs better when the sample size

is large, as theoretically analyzed in Gutmann and Hyvärinen (2012). In this sense, reducing the noise in embeddings is achieved by mean pooling from multiple embeddings which implicitly increases the sample size with respect to the random variable $\xi$ and potentially leads to improved performance, i.e., replacing $z_i^1$ and $z_i^2$ by $z_i^{1,-}$ and $z_i^{2,-}$ involving $K$ samples (in both positive pairs and negative pairs within Eq. (2)) through mean sampling may obtain better performance.

However, enlarging the sample size affects positive and negative pairs differently. As shown in Table 1, reducing noise in positive pairs through mean sampling results in unsatisfactory performance, while it improves performance in negative pairs. The main reason is that, under the SimCSE framework, the positive pairs require diversity as informative pairs for contrastive learning, which is reduced by mean sampling. Otherwise, the training signal in Eq. (2) may become trivial if there is no diversity between $z_i^1$ and $z_i^2$ for a positive pair, because $s(z_i^1, z_i^2) > s(z_i^1, z_j^2)$ when $z_i^1 = z_i^2$ and $i \neq j$. In summary, diversity is crucial for positive pairs, while minimizing noise is beneficial for negative pairs to achieve better performance.

### 3.2 Our Solution: Off-Dropout Sampling

Mean sampling significantly reduces the variance and yields better performance. However, $K$ times average sampling requires a time complexity overhead of $\mathcal{O}(KN)$.

To address this overhead, we propose off-dropout sampling, which turns off the dropout when sampling negative example representations. Off-dropout sampling produces representations with zero variance. At a high level, off-dropout sampling is empirically equivalent to the mean of infinite times resampling, as demonstrated by Hinton et al. (2012), which is also known as *weight scaling inference rule* (Goodfellow et al., 2016). Therefore, off-dropout sampling provides unbiased estimation of representation with zero variance, and

the sampling overhead is equal to that of default random sampling. Consequently, the InfoNCE objective for off-dropout sampling is:

$$\ell_{\text{off-Info}} = -\log \frac{e^{s(z_i^1, z_i^2)}}{e^{s(z_i^1, z_i^2)} + m \sum_{j=1, j \neq i}^{N} e^{s(z_i, z_j)}}$$

$$(3)$$

where $s(z_i, z_j)$ represents the similarity between negative pairs, and $z_i$, $z_j$ represents the representations sampled without dropout. $m$ is a trade-off factor between positive and negative examples.

It should be noticed that reducing the noise in negatives is very different from hyperparameter tuning: In principle, we investigate the sample size and thereby justify if the current sentence embedding methods satisfy the large sample size requirement from the NCE principle; In practice, tuning the dropout rate changes the distribution of dropout patterns, which violates the principle of controlling variables. Therefore, our strategy to reduce the noise in negatives is fundamentally different from parameter tuning in both principle and practice.

## 4 Mitigating Feature Corruption

**Feature Corruption Issue** Feature corruption [2] (Chen and He, 2021) illustrates the issue that each dimension of the output representation has high similarity with the other dimensions. Such correlation between dimensions reduces the model's representation capability and undermines downstream performance (Zbontar et al., 2021; Klein and Nabi, 2022).

### 4.1 Existing Solution

Zbontar et al. (2021) proposes BarlowTwins as an additive regulation to tackle such an issue, which is a dimension decorrelation objective. BarlowTwins tackles feature corruption by minimizing the redundancy between each dimension and aims to produce dimensional-independent representations. Formally, given a cross-correlation matrix $\mathbf{C} \in \mathbb{R}^{D \times D}$, its objective is:

$$\ell_{\text{BT}} = -\sum_{c}(1 - C_{cc})^2 + \lambda_{\text{BT}} \sum_{c} \sum_{d \neq c} C_{cd}^2$$

$$C_{cd} = \frac{\sum_i z_{i,c}^1 z_{i,d}^2}{\sqrt{\sum_i (z_{i,c}^1)^2} \sqrt{\sum_i (z_{i,d}^2)^2}}$$

$$(4)$$

Where $D$ is the total number of dimensions ($D$=768 for base model), $c$, $d$ are dimension indices, and $z_{i,c}^1$, $z_{i,d}^2$ are corresponding dimension values of the representation of the $i$-th sentence from a mini-batch of size $N$. However, such an objective does not yield gains over SimCSE when applied to sentence embeddings in STS tasks (Klein and Nabi, 2022).

### 4.2 Rank Bottleneck for BarlowTwins

BarlowTwins aims to achieve orthogonalization of all dimensions in the representation by maximizing the diagonal elements of the correlation matrix, denoted as $\mathbf{C} = (C_{cd})$, while minimizing the non-diagonal elements. In linear algebra, a parametrized matrix can be optimized to become an orthogonal matrix if there exists a parameter that ensures the matrix is of full rank. However, both theoretically and empirically, we observe that $\mathbf{C}$ is far from being a full-rank matrix, meaning its rank is close to $D$.

From a theoretical standpoint, if the denominator of $C_{cd}$ remains constant for any $c$ and $d$, $\mathbf{C}$ can be expressed as the product of a matrix with dimensions $D \times N$ and another matrix with dimensions $N \times D$. In this case, we can demonstrate that the rank of $\mathbf{C}$ is at most $\min(N, D)$. However, in the conventional settings of SimCSE, $N$ is 64 and $D$ is 768. [3] Consequently, the rank of $C$ is at most $N$, where $N \ll D$ for any parameter.

From an empirical perspective, we randomly sample a batch of 64 sentences and compute the rank of their cross-correlation matrix. We observe that the rank of the SimCSE correlation matrix is 64. Consequently, it is impossible to optimize a rank 64 matrix to become a rank 768 identity matrix using the BarlowTwins objective. The rank of the correlation matrix poses a bottleneck for the BarlowTwins objective, making it difficult to optimize $\mathbf{C}$ to become a full-rank matrix. Therefore, there is a rank bottleneck issue when optimizing the BarlowTwins objective. This might explain why BarlowTwins does not perform well when applied on top of SimCSE, as demonstrated in Table 2.

**Empirical Justification of the Rank Bottleneck** To verify the rank bottleneck hypothesis, one can adjust the batch size or reduce the total number of representation dimensions. However, increasing

---

[2]Feature corruption issue is also known as "feature/representation degeneration/collapse" problem, which originates from contrastive learning research in computer vision.

[3]Note that the batch size of 64 is the default setting in Sim-CSE, which achieved the best performance in both SimCSE original paper and our preliminary experiments.

| Objectives | STS12 | STS13 | STS14 | STS15 | STS16 | STS-B | SICK-R | Avg. |
|---|---|---|---|---|---|---|---|---|
| SimCSE-BERT$_{base}$ | 68.40 | **82.41** | **74.38** | 80.91 | **78.56** | 76.85 | **72.23** | **76.25** |
| +BarlowTwins ($D$=768) | 50.59 | 70.07 | 58.48 | 68.98 | 68.23 | 64.94 | 67.07 | 64.05 |
| +100 artificial $z$ | 66.19 | 77.62 | 67.67 | 75.17 | 73.34 | 72.37 | 67.26 | 71.37 |
| +300 artificial $z$ | 69.29 | 78.80 | 69.03 | 77.47 | 75.27 | 74.18 | 70.88 | 73.56 |
| +704 artificial $z$ | **70.52** | 81.86 | 73.72 | **81.17** | 76.95 | **77.21** | 71.10 | 76.08 |

Table 2: SimCSE performance with BarlowTwins additive objectives. We pad each mini-batch (batch size 64) embedding matrix with a group of artificial representations sampled from standard Gaussian distribution.

the batch size will alter the number of in-batch negatives, while reducing the representation dimensions will exacerbate the dimension bottleneck problem. Both methods will modify the default settings of SimCSE and consequently affect its performance.

To address this, we conduct a straightforward experiment without altering the SimCSE framework settings. We maintain the original SimCSE settings but introduce $M$ artificial embeddings to each mini-batch embedding matrix when calculating the BarlowTwins loss value. Thus, contrastive learning at the data level is still performed on $N$ batch size embeddings, while dimension-wise decorrelation is applied to the padded embedding matrix of size $N + M$. Consequently, we increase the rank of the correlation matrix by $M$ without modifying SimCSE.

We employ this approach to train the model, and the results are presented in Table 2. The table illustrates that the performance of the BarlowTwins objective improves as the number of padding artificial embeddings increases. By introducing these artificial embeddings, we successfully overcome the rank bottleneck issue of the correlation matrix.

### 4.3 Our Solution: Dimension-Wise Contrastive Learning

Previous experiments have confirmed the existence of the rank bottleneck issue in the BarlowTwins objective and have addressed this problem by padding artificial embeddings. However, optimizing parameters with a large number of artificial embeddings reduces training efficiency. Therefore, we propose a Dimension-wise Contrastive Learning (DCL) objective that naturally avoids the rank bottleneck issue. The DCL objective is defined as follows:

$$\ell_{DCL} = -\sum_{c=1}^{D} \log \frac{e^{s(z_{\cdot,c}^1, z_{\cdot,c}^2)}}{\sum_{d=1}^{D} e^{s(z_{\cdot,c}^1, z_{\cdot,d}^2)}} \quad (5)$$

The term $s(z_{\cdot,c}^1, z_{\cdot,d}^2)$ calculates the cross-dimension similarity between the $c$-th and $d$-th dimensions. We use dot product with batch normalization to measure similarity:

$$s(z_{\cdot,c}^1, z_{\cdot,d}^2) = \sum_i \tilde{z}_{i,c}^1 \tilde{z}_{i,d}^2 / \tau_{DCL}$$

$$\tilde{z}_{i,c} = \frac{z_{i,c} - \bar{z}_c}{\sigma_{z_c}}$$

Here, $\bar{z}_c = \frac{1}{N} \sum_i z_{i,c}$, $\sigma_{z_c}^2 = \frac{1}{N-1} \sum_i (z_{i,c} - \bar{z}_c)^2$.

The DCL objective represents dimension-wise contrastive learning. It improves upon the BarlowTwins objective in several ways: 1) Intuitively, Eq. 5 is a relative optimization that can be more easily optimized compared to the absolute regression objective (Gutmann and Hyvärinen, 2012); 2) This relative optimization avoids the rank bottleneck issue by only requiring the dimension to be relatively more "self-similar" compared to other dimensions, instead of necessitating a full-rank identity matrix as the only optimal solution.

By combining both proposed strategies with a trade-off factor $\lambda$, the final objective function for improving contrastive learning for sentence embeddings is as follows:

$$\ell = \ell_{off-info} + \lambda \ell_{DCL} \quad (6)$$

## 5 Experiments

### 5.1 Setups

**Baselines** We compare with several sentence representation methods on STS tasks, which includes GloVe embeddings (Pennington et al., 2014), Skip-thought (Kiros et al., 2015), BERT embeddings with pooling aggregation (Devlin et al., 2019), BERT-Flow(Li et al., 2020), and BERT-Whitening(Su et al., 2021).

We also compare with several recently proposed contrastive learning based sentence representation method, for instance, ISBERT (Zhang et al., 2020),

| Method | STS12 | STS13 | STS14 | STS15 | STS16 | STS-B | SICK-R | Avg. |
|---|---|---|---|---|---|---|---|---|
| GloVe(avg.) | 55.14 | 70.66 | 59.73 | 68.25 | 63.66 | 58.02 | 53.76 | 61.32 |
| BERT$_{base}$ (first-last avg.) | 39.70 | 59.38 | 49.67 | 66.03 | 66.19 | 53.87 | 62.06 | 56.70 |
| BERT$_{base}$ -flow | 58.40 | 67.10 | 60.85 | 75.16 | 71.22 | 68.66 | 64.47 | 66.55 |
| BERT$_{base}$ -whitening | 57.83 | 66.90 | 60.90 | 75.08 | 71.31 | 68.24 | 63.73 | 66.28 |
| IS-BERT$_{base}$ | 56.77 | 69.24 | 61.21 | 75.23 | 70.16 | 69.21 | 64.25 | 66.58 |
| CT-BERT$_{base}$ | 61.63 | 76.80 | 68.47 | 77.50 | 76.48 | 74.31 | 69.19 | 72.05 |
| ConSERT$_{base}$ | 64.64 | 78.49 | 69.07 | 79.72 | 75.95 | 73.97 | 67.31 | 72.74 |
| SCD-BERT$_{base}$ | 66.94 | 78.03 | 69.89 | 78.73 | 76.23 | 76.30 | **73.18** | 74.19 |
| SimCSE-BERT$_{base}$ | 68.40 | **82.41** | 74.38 | 80.91 | 78.56 | 76.85 | 72.23 | 76.25 |
| *SimCSE++-BERT$_{base}$ | **73.66** | 82.36 | **75.86** | **83.09** | **79.76** | **79.71** | 71.91 | **78.05** |
| *+Off-Info | 69.39 | 82.42 | 75.91 | 82.92 | 78.82 | 78.86 | 71.62 | 77.13 |
| *+DCL | 70.15 | 83.46 | 74.91 | 81.95 | 79.83 | 79.39 | 72.14 | 77.40 |
| ConSERT$_{large}$ | 70.69 | 82.96 | 74.13 | 82.78 | 76.66 | 77.53 | 70.37 | 76.45 |
| SimCSE-BERT$_{large}$ | 70.88 | 84.16 | 76.43 | 84.50 | **79.76** | 79.26 | 73.88 | 78.41 |
| *SimCSE++-BERT$_{large}$ | **72.37** | **85.37** | **78.68** | **84.69** | 79.57 | **80.37** | **74.05** | **79.30** |
| SBERT$_{base}$ | 70.97 | 76.53 | 73.19 | 79.09 | 74.30 | 77.03 | 72.91 | 74.89 |
| SimCSE-SBERT$_{base}$ | 69.41 | 80.76 | 74.37 | 82.61 | 77.64 | 79.92 | 76.62 | 77.33 |
| *SimCSE++-SBERT$_{base}$ | **72.92** | **83.45** | **77.19** | **83.46** | **79.38** | **81.54** | **76.79** | **79.25** |
| SBERT$_{large}$ | 72.27 | 78.46 | 74.90 | 80.99 | 76.25 | 79.23 | 73.75 | 76.55 |
| SimCSE-SBERT$_{large}$ | 76.16 | 83.77 | 77.27 | 84.33 | **79.73** | 81.67 | 77.25 | 80.03 |
| *SimCSE++-SBERT$_{large}$ | **76.66** | **84.76** | **78.53** | **84.37** | 79.66 | **82.37** | **78.09** | **80.63** |

Table 3: Main results on Semantic Textual Similarity benchmark dataset performance (Spearman correlation, "all" setting). Our proposed methods are marked with "*". The highest numbers among models with the same pre-trained encoder are highlighted. Off-dropout, DCL sampling and their combination - SimCSE++ - outperforms the baseline SimCSE with $p < 0.005$.

CT-BERT (Carlsson et al., 2021), ConSERT (Yan et al., 2021), together with the current mainstream SimCSE (Gao et al., 2021) and SOTA DiffCSE (Chuang et al., 2022).

**Dataset** We use the default one million randomly sampled sentences from English Wikipedia for unsupervised training, as previous studies (Gao et al., 2021; Chuang et al., 2022; Zhang et al., 2022c; Wu et al., 2022) are all conducted on this corpus. We do not conduct any data selection or sampling strategy during the training.

**Evaluation** We evaluate our model on 7 sentence semantic textual similarity (STS) tasks, which includes STS tasks 2012-2016 (Agirre et al., 2012), STS Benchmark (Cer et al., 2017), and SICK-Relatedness (Marelli et al., 2014). We follow Sim-CSE (Gao et al., 2021) settings of MLP layers, and employ MLP on top of [CLS] token representation for training while removing MLP for evaluation. We evaluate the model for every 125 updating steps based on the STS-B development set, without any gradient accumulation. And evaluate the best

checkpoint at the final evaluation on test sets.

**Implement Details** We conduct the experiments using pre-trained checkpoints from BERT$_{base}$ and BERT$_{large}$ (Devlin et al., 2019) with Huggingface Transformer (Wolf et al., 2020) framework. Besides, to illustrate the compatibility of SBERT, following Zhang et al. (2022c) settings, we also employ SBERT (Reimers and Gurevych, 2019) on NLI (Conneau et al., 2017; Reimers and Gurevych, 2019) variant checkpoints for experiments.

During the training, the contrastive temperature $\tau$ is the same as SimCSE to be $0.05$. And the trading-off ratio $m$ is set as $0.9$. For DCL, we set temperature $\tau_{DCL}$ as $5$ and loss coefficient $\lambda$ as $0.1$. We train the model for one epoch with a learning rate $3e^{-5}$ for base model and $8e^{-6}$ for the large model with the same batch size $64$ and sequence length $32$. The model is optimized by Adam (Kingma and Ba, 2014) optimizer with default settings without gradient accumulation.

| Objectives | STS12 | STS13 | STS14 | STS15 | STS16 | STS-B | SICK-R | Avg. |
|---|---|---|---|---|---|---|---|---|
| BERT$_{base}$ (first-last avg.) | 39.69 | 59.37 | 49.67 | 66.03 | 66.19 | 53.88 | 62.06 | 56.70 |
| +whitening(wiki) | 45.64 | 64.38 | 56.57 | 70.35 | 68.64 | 60.32 | 63.42 | 61.33 |
| +DCL(wiki,$\tau_{DCL}$=0.05) | 52.96 | 73.85 | 62.80 | 72.12 | 71.25 | 68.15 | **68.36** | 67.07 |
| +flow (NLI) | 58.40 | 67.10 | 60.85 | 75.16 | 71.22 | 68.66 | 64.47 | 66.55 |
| +whitening(NLI) | 57.83 | 66.90 | 60.90 | 75.08 | 71.31 | 68.24 | 63.73 | 66.28 |
| +DCL(NLI, $\tau_{DCL}$=0.05) | **59.25** | **74.04** | **63.91** | **75.85** | **72.46** | **70.67** | 67.05 | **69.03** |
| SimCSE-BERT$_{base}$ | **68.40** | **82.41** | **74.38** | **80.91** | **78.56** | **76.85** | **72.23** | **76.25** |
| +whitening | 59.27 | 76.68 | 67.22 | 74.86 | 73.85 | 68.43 | 69.22 | 69.93 |
| +flow | 61.33 | 78.54 | 69.87 | 77.47 | 76.02 | 71.73 | 70.70 | 72.24 |
| DiffCSE (Chuang et al., 2022) | 72.28 | **84.43** | 76.47 | **83.90** | 80.54 | 80.59 | 71.23 | 78.49 |
| DiffCSE (reproduced)[1] | 69.41 | 82.47 | 74.52 | 82.82 | 80.06 | 79.39 | 72.05 | 77.25 |
| *+SimCSE++ | **72.68** | 83.40 | **76.76** | 83.66 | **80.57** | **80.94** | **72.82** | **78.69** |

Table 4: Block 1: DCL compared with post-processing methods. NLI is used without labels; Block 2: Post-processing methods on top of SimCSE lead to unsatisfying performance; Block 3: SimCSE++ is robust to the non-SimCSE framework. 1: Using officially released source code, and our method improves its performance with $p < 0.005$.

## 5.2 Main Results

The evaluation results are shown in Table 3, SimCSE++ outperforms the previous approaches. Compared with its baselines, our methods advance the average Spearman correlation coefficient from 76.25% to 78.05%, and its large variant raises the average Spearman correlation score further to 79.30%. SimCSE++ also improves the SBERT variants' performances. Compared with SimCSE, SimCSE++ achieves 79.25% and 80.63% on base and large variants, which shows an even stronger representation ability.

We also explore the contribution of DCL objective and off-dropout sampling in Table 3. It shows that the off-dropout sampling strategy alone is able to improve the sentence semantic representation to 77.13% Spearman correlation score, and DCL objective with normal dropout augmented negative contrastive term achieves 77.40%.

## 5.3 Ablation Study

We investigate the effect of the hyperparameters on the whole system on the STS-B development set of BERT$_{base}$ model in Table 5. We search $m$ in the range $\{0.5, 0.8, 0.9, 1, 1.1, 1.2\}$. The optimal value is 0.9. We search the aggregation weight $\lambda$ for DCL within the range $\{0.02, 0.05, 0.1, 0.2, 0.5, 1\}$, and the optimum value is 0.1. We carry out the DCL temperature search in ranging in $\{1, 2, 5, 10, 20, 50\}$, and optimal DCL temperature is 5.

Following Gao et al. (2021), we also plot $\ell_{align}$-$\ell_{uniform}$ joint plot at Appendix A. Further, we con-

| $m$ | 0.5 | 0.8 | 0.9 | 1 | 1.1 | 1.2 |
|---|---|---|---|---|---|---|
| STS-B dev | 68.55 | 81.61 | **83.77** | 79.11 | 79.99 | 71.49 |
| $\lambda$ | 0.02 | 0.05 | 0.1 | 0.2 | 0.5 | 1 |
| STS-B dev | 82.60 | 83.25 | **83.77** | 82.36 | 80.79 | 77.79 |
| $\tau_{DCL}$ | 1 | 2 | 5 | 10 | 20 | 50 |
| STS-B dev | 82.16 | 82.43 | **83.77** | 83.56 | 83.37 | 81.76 |

Table 5: Searching for weight term $m$, DCL objective weight $\lambda$ and DCL temperature $\tau_{DCL}$ on STS-B development set.

| Model | Training Time |
|---|---|
| SimCSE-BERT$_{base}$ | 1h 50 min |
| *SimCSE++-BERT$_{base}$ | 1h 59 min |

Table 6: 1 epoch training time for SimCSE and our proposed SimCSE++

duct the qualitative comparison on sentence retrieval tasks in Appendix B to further illustrate our improvement.

**Comparing with post-processing** We compare the single DCL objective with widely applied post-processing methods (i.e. whitening and flow model). Table 4 shows that the DCL objective outperforms all the post-processing methods.

**Robustness to other framework** In Table 4, we introduce our method into the DiffCSE framework using officially released source code with our proposed methods. As a result, we further advance DiffCSE baseline by 1.4 points based on our reproduced results.

**Runtime Efficiency** We compare the training time between SimCSE and our proposed Sim-CSE++. It shows that, the off-dropout sampling, and DCL do not introduce noticeable running time overhead compared to the SimCSE baseline. Moreover, we observe that both SimCSE and our proposed SimCSE++ converge to their optimum within the first 5k training steps, which is around 30 minutes of training. Consequently, the overhead of our modification is negligible.

## 6 Conclusion

In this paper, we improve CL-based sentence embeddings in dropout noise and feature corruption. The main findings are: 1) having modest dropout noise is successful for positive pairs and reducing dropout noise from positive pairs is harmful whereas reducing dropout noise from negative pairs is beneficial; 2) the well-known solution to feature corruption does not lead to gains on sentence embedding due to the rank bottleneck issue. Accordingly, we propose off-dropout to eliminate the dropout randomness from negative pairs and dimension-wise CL objective to break the bottleneck to alleviate feature corruption, both of which outperform strong baselines by a margin.

## Ethical Considerations

This study focuses on the representation of sentences, the objective of which is to achieve better performance on general domain sentence similarity tasks. Therefore, the training corpus and benchmark datasets are open source and do not contain any personally sensitive information; And we employ widely applied pre-trained language models with commonly used contrastive learning strategies, thereby having no impact on the political, social, or natural environment.

## Limitations

The limitations consist of two aspects: for dropout noise, a novel sampling strategy for positive pairs is left unexplored; for DCL, it could be improved by applying more advanced data-wise contrastive learning strategies.

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

| | Unsup-SimCSE | SimCSE++ |
|---|---|---|
| **Query1: An animal is biting a persons finger.** | | |
| #1 | A dog is biting a twig. | A dog bites someone 's finger. |
| #2 | A dog bites someone 's finger. | A dog is biting a twig. |
| #3 | Small black dog biting on a person 's finger. | Small black dog biting on a person 's finger. |
| #4 | A dog is biting a mop. | The dog is biting a stick. |
| #5 | A dog biting a man 's rear | A dog biting at its own rear. |
| **Query2: A man plays the violin.** | | |
| #1 | A woman plays the violin. | A man playing the violin. |
| #2 | A man plays the violin on stage. | A man plays the violin on stage. |
| #3 | A man playing the violin. | A woman plays the violin. |
| #4 | A musician playing his violin. | A sitting man playing the violin. |
| #5 | A man plays a violin while smiling. | A musician playing his violin. |

Table 7: Retrieved top-5 examples by SimCSE and SimCSE++ from Flickr30k (150k sentences).

## A  Alignment and Uniformity

As illustrated by Wang and Isola (2020), models have both good alignment and uniformity and usually achieve better performance. Alignment is a measure for representation consistency of the same input instance:

$$\ell_{\text{align}} = \mathop{\mathbb{E}}_{(x^1, x^2) \sim p_{\text{pos}}} \|f(x^1) - f(x^2)\|^2 \quad (7)$$

Since we adopt dropout as augmentation, i.e. $x = x^+$ and the only difference is the dropout pattern $\epsilon^1$ and $\epsilon^2$. And $p_{\text{pos}}$ indicate the positive pairs are sampled from positive datasets. Uniformity is a measure for representation distribution on representation space, which is defined by:

$$\ell_{\text{uniform}} = \log \mathop{\mathbb{E}}_{x,y \overset{i.i.d}{\sim} p_{\text{data}}} e^{-2\|f(x) - f(y)\|^2} \quad (8)$$

Where $p_{\text{data}}$ denotes whole data distribution, and $x$ and $y$ are instance randomly sampled from dataset. Fig 1 shows the $\ell_{\text{align}}$-$\ell_{\text{uniform}}$ joint plot, following Gao et al. (2021).

## B  Qualitative Comparison

We also conduct the retrieval comparison between SimCSE++ and its baseline SimCSE on Flickr30k dataset (Young et al., 2014). We use 150k captions from Flickr30k for images and take any random sentence as query to retrieve similar sentences (based on cosine similarity measure). Examples is shown in Table 7. We find that the retrieved sentences from SimCSE++ is of higher quality compared to those retrieved from SimCSE: SimCSE++ retrieves better top-1 sentences with most similar

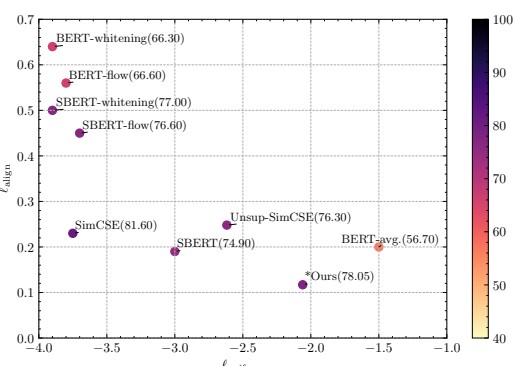

Figure 1: $\ell_{\text{align}}$-$\ell_{\text{uniform}}$ plot of base model.

semantic information for Query@1, and preserves the correct gender information on third person pronoun in #1 for Query@2.