# OpenReview forum: "SimCSE++: Improving Contrastive Learning for Sentence Embeddings from Two Perspectives"
_EMNLP/2023/Conference — EMNLP 2023 Main_

### Official Review · Reviewer_RiEf · 2023-07-28

**Soundness:** 4

**Excitement:**

4: Strong: This paper deepens the understanding of some phenomenon or lowers the barriers to an existing research direction.

**Paper Topic And Main Contributions:**

This paper propose to improve the SimCSE method from two perspectives, namely handling dropout noise and addressing feature corruption. They propose two strategies, namely off-dropout and dimension-wise contrastive learning, to alleviate the dropout noise and feature corruption.
Experiment results on several benchmark datasets verify the effectiveness of this method.
This work also provides in-detail analysis for the proposed strategies.

**Questions For The Authors:**

Can you make a comparison with the text embedding APIs provided by OpenAI?( https://platform.openai.com/docs/api-reference/embeddings)

**Reasons To Accept:**

1. This work propose two important issues which have long existed in the Contrastive Learning-based representation but ignored by previous work.
2. The proposed two strategies are simple yet effective, leading to improvement on several benchmark datasets;
3. The authors provide in-detail analysis for their methods which are insightful and inspirational;


**Reasons To Reject:**

Only experimenting on 1 million randomly sampled sentences and BERT-base/large architecture. In the context of large-language model, the scalability of the proposed method are not mentioned.

**Reproducibility:**

4: Could mostly reproduce the results, but there may be some variation because of sample variance or minor variations in their interpretation of the protocol or method.

**Reviewer Confidence:**

3: Pretty sure, but there's a chance I missed something. Although I have a good feel for this area in general, I did not carefully check the paper's details, e.g., the math, experimental design, or novelty.

---

> ### Author Rebuttal · Authors · 2023-08-29
>
> ### R1: Only experimenting on 1 million corpus and BERT-base/large architecture; the scalability of the proposed method
> **[1M wiki corpus is the default data setting typical in this field]** The wiki 1 million data used in this paper originated from SimCSE paper, and many proposed methods will utilize this corpus for experiments. Here we directly inherit this corpus for a fair comparison with previous methods.
>
> **[SBERT]** Besides BERT structure, we also introduce SBERT structure in the experiment sections in the paper, which follows previous studies [1].
>
> **[Scalability]** In principle, our proposed method is scalable to data scale (batch size) and model size (BERT-base to BERT-large), and our experiment shows that performance increases with the model size. Therefore, our method is generic to scalability to LLMs. However, fine-tuning LLMs is difficult but could be tackled by some specific technique, which remains as future work.
>
> ### Q1:  comparison with the text embedding APIs provided by OpenAI
> **[On par performance to LLMs but low training cost and inference overhead]** Obtaining the sentence embeddings from large language models, i.e. Chat-gpt and GPT-4 is impossible since Open-AI does not provide such an API for model logits.
> However, they provide the embedding API from “text-embedding” series model and report the corresponding performance (please refer to the link: https://openai.com/blog/new-and-improved-embedding-model). Their average performance for STS tasks is around 80.5 on STS12-16 testing set, and such a performance is comparable to our large model.
> Moreover, compared to them, our method is of significantly low training cost. Besides, since our model size is much smaller than LLMs, we have a significantly shorter inference overhead.
>
> ### Reference
> [1] Yuhao Zhang, Hongji Zhu, Yongliang Wang, Nan Xu, Xiaobo Li, and Binqiang Zhao. 2022. A Contrastive Framework for Learning Sentence Representations from Pairwise and Triple-wise Perspective in Angular Space. In Proceedings of the 60th Annual Meeting of the Association for Computational Linguistics (Volume 1: Long Papers), pages 4892–4903, Dublin, Ireland. Association for Computational Linguistics.

---

### Official Review · Reviewer_Amxx · 2023-08-04

**Soundness:** 4

**Excitement:**

4: Strong: This paper deepens the understanding of some phenomenon or lowers the barriers to an existing research direction.

**Paper Topic And Main Contributions:**

This paper identifies two issues with SimCSE and proposes two solutions.

First, the application of dropout has not been thoroughly investigated. SimCSE only investigated the use of standard dropout as noise and only examined the influence of dropout rate. This paper further investigates the effect of dropout noise level by adding different amounts of noise to the positive and negative samples. They find that while adding some level of noise to the positive samples is beneficial, adding dropout noise to negative samples is purely harmful to performance. Therefore, they propose using off-sample for dropout noise. They avoid adding noise to the negative samples and maintain the standard dropout noise for positive samples.

Second, the authors argue that SimCSE has an issue with feature corruption. They describe the issue as the high similarities in dimensions of a model's representation. They propose addressing this issue by applying the recently proposed Barlow Twins method.

By combining these two methods, they are able to significantly increase the performance of SimCSE. The two methods are also compatible with newer SimCSE-based methods, such as DiffCSE.

**Questions For The Authors:**

Is _feature corruption_ the commonly used name for the issue?  This name of the issue is not mentioned by Zbontar et al. (2021).

Will you release the source code upon acceptance?

**Reasons To Accept:**

I would like to extend my congratulations to the authors for successfully identifying two commonly overlooked aspects of SimCSE and for providing compelling solutions to address these issues. The authors have effectively presented the two main problems of SimCSE along with empirical evidence to support their arguments. Moreover, the suggested solutions are meticulously explained in great detail. Undoubtedly, the findings of this study will greatly benefit the research community.

**Reasons To Reject:**

Unfortunately, this paper is not well written, as there are numerous typos and grammatical errors throughout. Additionally, there is an unfinished sentence present (caption of table 1). Consequently, reading and comprehending the paper becomes quite challenging.

Moreover, there are also numerous imprecise uses of terminology. These misuses create confusion while reading and can hinder future work.
- [241] Eq. (2) involves 2xN _BERT structures_.  ---> BERT representations
- [249] we respectively add more noise (+Noise) or reduce some noise (-Noise) from z...
-> we respectively add **more** noise (+Noise) or **less** noise (-Noise) **to** z...
- Is _feature corruption_ the commonly used name for the issue?  This name of the issue is not mentioned by Zbontar et al. (2021).

The authors also did not mention whether the source code will be released publicly.  This will make the reproduction process much more difficult.

**Reproducibility:**

2: Would be hard pressed to reproduce the results. The contribution depends on data that are simply not available outside the author's institution or consortium; not enough details are provided.

**Reviewer Confidence:**

5: Positive that my evaluation is correct. I read the paper very carefully and I am very familiar with related work.

**Typos Grammar Style And Presentation Improvements:**

This paper has several grammatical mistakes.  Here are some common ones:
- Typically a comma is placed after an in-sentence equation, and the following clause explaining the equation should start with the lower case.  Take equation (3) as an example:
$$ \ell = ... (3), $$ **w**here $s$...
- A space should be added between a bracket and its preceding word:
Just to name a few occurrences:

[503] as previous studies(Gao et al., 2021; ...

[510] 2021-2016(Agirre et al., 2012)

- Please double check capitalisation.  The first letter of each sentence should be capitalised.

[286] ... performance. 2) **t**he model...

[264] variable $\xi$. and thus... -> the period should be a comma.

One-off typos:
- [266] missing . in ... Eq (2) ...
- [259] extra . after footnote mark 1.
- [083] Need a space after the bracket in "...learning (DCL)to break"
- [265+] Missing period in the sentence in footnote 1.


Please proof read the paper more carefully upon acceptance.  The aforementioned typos and mistakes are by no means exhaustive.

---

> ### Author Rebuttal · Authors · 2023-08-29
>
> ### R1:  Imprecise uses of terminology.
> Thanks for the valuable feedback, we will polish the sentences and revise the terminology and equations for clarity, and we will release the source code once the paper is accepted.
>
> ### Q1: Is feature corruption the commonly used name for the issue?
>
> Feature corruption, also known as *“feature/representation degeneration/collapse”*, is one of the fundamental problems in contrastive learning. Such a question originates from contrastive learning in images [1]. Since the studies for feature corruption are many and mainly in images, here, we only cite the paper that directly tries to address such an issue. We will add the footnote here to explain the feature corruption phenomenon for better understanding.
>
> ### Q2: release the source code
> Yes, we will release the source code with checkpoints and add the link to the content of the paper when it is published.
>
> ### Reference
> [1] Chen, Xinlei, and Kaiming He. "Exploring simple siamese representation learning." Proceedings of the IEEE/CVF conference on computer vision and pattern recognition. 2021.

---

### Official Review · Reviewer_3KpX · 2023-08-04

**Soundness:** 4

**Excitement:**

4: Strong: This paper deepens the understanding of some phenomenon or lowers the barriers to an existing research direction.

**Paper Topic And Main Contributions:**

This work first investigates the potential weaknesses in the common practices used in the self-supervised sentence embedding framework, SimCSE. First, they point to the dropout used in the contrastive loss. Through empirical investigation, the paper finds the noise to be useful mostly for positive pairs. Secondly, the paper addresses the "Corruption Issue" where the components of the representations become highly correlated. Typically, BarlowTwins objective is added to mitigate this issue. However, the paper finds an intrinsic issue (rank bottleneck) with the this method. For both of these problems, the paper proposes simple solutions that is shown to be effective in improving the SOTA. The training is done a subset of wikipedia sentences (typical in this field) and the evaluation is done on sentence similarity tasks (STS).

**Questions For The Authors:**

Refer to the weaknesses section.



**Reasons To Accept:**

A1. The paper shed light on the potential issues with a common sentence embedding representation algorithm. This itself will allow future work to improve the framework or design better algorithms. Also, the investigation done by the paper provide valuable insights into the inner working of such algorithms.

A2. Although I'm more excited about the empirical investigation part of the paper, the proposed solutions achieve improve upon the original SimCSE, moreover the these solutions seem to be general and can be applied as an auxiliary objectives on top other algorithms.

A3. The evaluation setup includes many baselines which helps to put the results in the context.

**Reasons To Reject:**

I didn't find any major weakness with this paper. But the following could be used to improve the work. *though this is not my main area of expertise and I'm not very familiar with all the work in this line.*

R1. The paper does not provide any runtime efficiency metrics for SimCSE++. It seems there are many reduction operation happening in the DCL objective. It's not clear if training with DCL objective is slower or not. If it's slower then one could argue that the time spend on computing DCL objective can be instead used to train on more data/iteration which can further improve the performance.

R1. One limitation of the evaluation setup could be the limited size of the training and model sizes. It's convincible that with enough compute and data the issue tackled by the paper be mitigated or not be contributing to much.

R2. The clarity of presentation in the paper can be improved. The mathematical notation is hard to follow especially in lines 262 and 468.

**Reproducibility:**

4: Could mostly reproduce the results, but there may be some variation because of sample variance or minor variations in their interpretation of the protocol or method.

**Reviewer Confidence:**

3: Pretty sure, but there's a chance I missed something. Although I have a good feel for this area in general, I did not carefully check the paper's details, e.g., the math, experimental design, or novelty.

**Typos Grammar Style And Presentation Improvements:**

Line 254 second $z_i^{1,+}$ -> $z_i^{2,+}$

Line 267: $\langle z_i^1, z_i^1 \rangle$ -> $\langle z_i^1, z_i^2 \rangle$

---

> ### Author Rebuttal · Authors · 2023-08-29
>
> ### R1: runtime efficiency
> **[Negligible overhead]**: According to our experiments, the proposed methods, i.e. off-dropout sampling, and DCL do not introduce noticeable running time overhead compared to the SimCSE baseline. The table below shows the training time for 1 epoch iteration on the Wiki1m corpus.
>
> | **Model** | **Training Time** |
> |---|---|
> | SimCSE | 1h 50 min |
> | SimCSE++ | 1h 59 min |
>
> **[More data will not achieve better results]** Moreover, under unsupervised settings, both SimCSE and our proposed SimCSE++ converge to the optimum within the first 5k training steps, which is around 30 minutes of training. Since SimCSE is sensitive to batch size (batch size 64 gives the best performance), as reported in its original paper and our experimental observations, increasing more data will not contribute to model performance.
>
> ### R2:  limited size of the training and model sizes
> **[Our method also improves the large model baseline performance]** As illustrated in the previous studies, more data will not achieve better results due to the unsupervised learning nature [1]. Indeed, increasing the model scale contributes to the performance, as demonstrated in our experimental results. In particular, when applying the proposed method on BERT-large, its performance still increases with the model scale (78.05% for SimCSE++-BERT-base to 79.30% for SimCSE++-BERT-large), which illustrates the contribution of our method is orthogonal to the contribution of the model scale. Therefore, we think that our method is effective when continuously enlarging the model size.
>
> ### R3: The clarity of presentation
> Thanks for the feedback, we will revise the equations for clarity.
>
> ### Reference
> [1] Tianyu Gao, Xingcheng Yao, and Danqi Chen. 2021. SimCSE: Simple Contrastive Learning of Sentence Embeddings. In Proceedings of the 2021 Conference on Empirical Methods in Natural Language Processing, pages 6894–6910, Online and Punta Cana, Dominican Republic. Association for Computational Linguistics.

---

### Meta-Review · Area_Chair_v6kq · 2023-09-18

**Recommendation:** 5

**Metareview:**

This is a strong paper on improving sentence representations. The authors start with "standard" contrastive learning and present two improvements. The paper presents an intuitive rationale and solid empirical results.

Ideally the paper would have been more polished at submission time. I don't see any reason to believe that the authors will have issues fixing the wording suggestions raised by the reviewers (and thank you to the reviewers for the thorough reviews!).

The paper would benefit from an analysis of why the method works. Solid empirical results are certainly a strength of the paper, but I recommend the authors explore when the method works better than the "standard" contrastive learning. This analysis will be limited by the choice of task (only sentence similarity), but I think it would be interesting. Does their method perhaps work better when syntactic structure is different? Is it better at subject replacements?

---

### Decision · Program_Chairs · 2023-10-07

**Decision:**

Accept-Main

**Comment:**

This is a strong paper on improving sentence representations. The authors start with "standard" contrastive learning and present two improvements. The paper presents an intuitive rationale and solid empirical results.

Ideally the paper would have been more polished at submission time. I don't see any reason to believe that the authors will have issues fixing the wording suggestions raised by the reviewers (and thank you to the reviewers for the thorough reviews!).

The paper would benefit from an analysis of why the method works. Solid empirical results are certainly a strength of the paper, but I recommend the authors explore when the method works better than the "standard" contrastive learning. This analysis will be limited by the choice of task (only sentence similarity), but I think it would be interesting. Does their method perhaps work better when syntactic structure is different? Is it better at subject replacements?